# Manipulating the Subcellular Localization and Anticancer Effects of Benzophenothiaziniums by Minor Alterations of N-Alkylation

**DOI:** 10.3390/molecules28041714

**Published:** 2023-02-10

**Authors:** Yanping Wu, Yuncong Chen, Shankun Yao, Shumeng Li, Hao Yuan, Fen Qi, Weijiang He, Zijian Guo

**Affiliations:** 1State Key Laboratory of Coordination Chemistry, Coordination Chemistry Institute, School of Chemistry and Chemical Engineering, Nanjing University, Nanjing 210023, China; 2Chemistry and Biomedicine Innovation Center (ChemBIC), Nanjing University, Nanjing 210023, China

**Keywords:** benzophenothiaziniums, type I photosensitizers, reactive oxygen species, phototoxicity index, lysosomal membrane permeabilization, mitochondria membrane potential, nicotinamide adenine dinucleotide phosphate, photodynamic therapy

## Abstract

Cationic, water-soluble benzophenothiaziniums have been recognized as effective type I photosensitizers (PSs) against hypoxic tumor cells. However, the study of the structure–property relationship of this type of PS is still worth further exploration to achieve optimized photodynamic effects and minimize the potential side effects. Herein, we synthesized a series of benzophenothiazine derivatives with minor N-alkyl alteration to study the effects on the structure–property relationships. The cellular uptake, subcellular organelle localization, reactive oxygen species (ROS) generation, and photocytotoxicity performances were systematically investigated. NH_2_NBS and EtNBS specifically localized in lysosomes and exhibited high toxicity under light with a moderate phototoxicity index (PI) due to the undesirable dark toxicity. However, NMe_2_NBS with two methyl substitutions accumulated more in mitochondria and displayed an excellent PI value with moderate light toxicity and negligible dark toxicity. Without light irradiation, NH_2_NBS and EtNBS could induce lysosomal membrane permeabilization (LMP), while NMe_2_NBS showed no obvious damage to lysosomes. After irradiation, NH_2_NBS and EtNBS were released from lysosomes and relocated into mitochondria. All compounds could induce mitochondria membrane potential (MMP) loss and nicotinamide adenine dinucleotide phosphate (NADPH) consumption under light to cause cell death. NMe_2_NBS exhibited remarkable in vivo photodynamic therapy (PDT) efficacy in a xenograft mouse tumor (inhibition rate, 89%) with no obvious side effects. This work provides a valuable methodology to investigate the structure–property relationships of benzophenothiazine dyes, which is of great importance in the practical application of PDT against hypoxia tumor cells.

## 1. Introduction

Photodynamic therapy (PDT) is a light-excited chemotherapy, which is a non-invasive clinical procedure to treat tumors after surgery, chemotherapy, and radiotherapy [1]. This technology could specifically kill tumor cells or tissues by generating reactive oxygen species (ROS) in the presence of a laser, oxygen, and a photosensitizer (PS) [2]. PS plays a crucial role in PDT processes, which can be classified into type I (electron transfer) and type II (energy transfer). At present, the most common clinical application is to use the type II mechanism to produce singlet oxygen to achieve the PDT effect. However, the mechanism dependence on oxygen content hinders the treatment of the hypoxia microenvironment in solid tumor tissue. The type I process generating free radicals and ionic radicals exhibited significant anti-tumor effects in normoxia and hypoxia conditions because of its diminished O_2_-dependence [2]. Nowadays, many PSs have been developed and investigated, including porphyrins, chlorines, phthalocyanines, cyanines, rhodamines, BODIPYs, and phenothiaziniums [3]. Among them, phenothiaziniums have received much attention due to their good water solubility, long wavelengths, and large molar extinction coefficients. Methylene blue (MB) is the most representative phenothiazinium with good solubility and a high singlet oxygen quantum yield; it has been used in conventional PDT antimicrobial research and PDT antitumor studies [4,5,6,7]. 

The phenothiazinium-fused benzene ring forms benzophenothiaziniums, possessing a tetracyclic scaffold. Approximately twenty years ago, James W. Foley et.al. reported Nile blue derivatives with oxygen atoms substituted by sulfur or selenium, which posse increased singlet oxygen [8]. Subsequently, Conor L. Evans reported benzophenothiazinium EtNBS against ovarian cancer cells in a 3D tumor model through both type I- and type II-mediated phototherapy [9,10]. In recent years, Peng and coworkers reported sulfur-substituted benzophenoxazine derivatives as powerful O_2_^−•^ radical generators that can eliminate hypoxic tumors via type I photoreaction [11,12,13]. Additionally, previous studies showed lysosomal localization of benzophenothiaziniums and photoinduced cell death via the lysosome disruption pathway [14]. Benzophenothiaziniums could lead to undesirable dark toxicity by affecting the integrity of the lysosomal membrane [15]. However, the minor structural modification effects of benzophenothiaziniums on the mechanism of PDT and its cytotoxic mechanisms remain unclear.

Herein, we synthesized three benzophenothiazine analogs NH_2_NBS, EtNBS, and NMe_2_NBS hexafluorophosphate with different substitutions of primary, secondary, and tertiary amines to keep the basic photophysical properties similar through minimal structural changes (Figure 1). Our results indicated that these NIR cationic PSs produced ROS via both type I and type II pathways. All of the compounds could be rapidly taken up by cancer cells; NH_2_NBS and EtNBS are mainly localized in lysosomes while NMe_2_NBS is located more in mitochondria than in lysosomes. We found that all three PSs exhibited high phototoxicities in normoxia and hypoxia. However, NH_2_NBS and EtNBS were more toxic in the dark and light, while two methyl groups that modified NMe_2_NBS displayed excellent phototherapeutic indices with significantly lower dark toxicity in HepG2 cells. We observed the migration of NH_2_NBS and EtNBS from lysosomes to mitochondria, concomitant with lysosomal membrane permeabilization (LMP) and mitochondria membrane potential (MMP) loss under irradiation. Of note, NMe_2_NBS possessed remarkable photodynamic antitumor therapy efficacy in a xenograft mouse tumor with good biocompatibility. To the best of our knowledge, this is the first report where alkylated benzophenothiazine induced cell death via destroying lysosomes and mitochondria, reducing NADPH levels under near-infrared light irradiation.

## 2. Results and Discussion

### 2.1. Synthesis and Characterizations of NH_2_NBS, EtNBS, and NMe_2_NBS

Three NIR cationic photosensitizers NH_2_NBS, EtNBS, and NMe_2_NBS, were synthesized according to procedures previously described [16]. The synthesis routes are shown in Figure 2. First, we synthesized Bunte salt via the Friedel−Crafts acylation. Then, benzophenothiaziniums were constructed by condensation. Third, hexa-fluorophosphate replaced chloride ion. All complexes were subsequently characterized by ^1^H NMR, ^13^C NMR, and HRMS (Appendix A). Detailed synthesis methods and characterizations are described in the section on experiments.

### 2.2. Photophysical Properties

The photophysical properties of three cationic photosensitizers were determined in several solvents. The main absorption and emission bands of the three compounds were in the ranges of 600–800 nm, which were in the therapeutic window. In the dichloromethane solution, all complexes are monomers with maximum absorptions at 642, 651, and 676 nm for NH_2_NBS, EtNBS, and NMe_2_NBS, respectively. Due to the different alkylation modifications, the absorption and emission wavelengths exhibited redshifts. The detailed photophysical data of the three compounds are displayed in Appendix A and Figure 1a,b and Appendix A.

### 2.3. ROS Generation Analysis

Since the photoinduced ROS is the main cytotoxic substance in the PDT process, we detected singlet oxygen (type II PDT) and superoxide radical (type I PDT) generation of three cationic photosensitizers in the solution. First, we measured the singlet oxygen quantum yield (Φ_Δ_) by the reference method using methylene blue (MB) as the standard reference and DPBF as a ^1^O_2_ scavenger in methanol [17]. As shown in Figure 1c, three compounds caused DPBF to bleach under irradiation for 30 s (635 nm laser irradiation). The values of the singlet oxygen quantum yield for NH_2_NBS, EtNBS, and NMe_2_NBS were 0.057, 0.060, and 0.034 (Appendix A). Second, the superoxide radical generations of three photosensitizers were determined by using the commercial O_2_^−•^ fluorescent probe dihydrorhodamine 123 (DHR123) in the PBS solution (Appendix A). As shown in Figure 1d, NH_2_NBS, EtNBS, and NMe_2_NBS significantly increased the fluorescence intensity of DHR123 at 526 nm under NIR light irradiation for 6 min. The results indicated that all of the compounds are efficient O_2_^−•^ radical generators. Thus, we speculated that NH_2_NBS, EtNBS, and NMe_2_NBS produce ROS through both type I and II mechanisms during the photodynamic therapy process, which can alleviate the oxygen dependency of photosensitizers to some extent.

### 2.4. Cell Uptake Studies

The cell uptakes of three PSs were measured at different time intervals by a UV–Vis spectrophotometer [18]. The quantitation of cellular PSs can be calculated by absorbance values. The cell uptake assay revealed that these PSs entered cells rapidly and the concentrations of intracellular PSs increased gradually (Appendix A). The cell uptake trend was NH_2_NBS > EtNBS > NMe_2_NBS at different time intervals. After 60 min of incubation with these PSs, approximately 40% of PSs were taken up by HepG2 cells, which is consistent with the reported phenothiazinium photosensitizers [18]. Additionally, we found that the cell uptake of NMe_2_NBS in a normal human hepatocyte L02 cell line is lower than the HepG2 cells (Appendix A). Several studies indicated that cationic dyes located in mitochondria tend to rely on the membrane potential for their uptake, while cationic dyes located in lysosomes tend to rely on the pH gradient penetrating the lysosome membrane for uptake [19]. Precious research studies have proved that the uptake mechanism of benzophenothiazine depends on the membrane’s pH gradient. NH_2_NBS and EtNBS are mainly localized in lysosomes, while NMe_2_NBS are mainly localized in mitochondria. We conjecture that the uptake of NMe_2_NBS can be modulated by membrane potential. As we all know, cancer cells show higher mitochondrial membrane potential than normal cells [20]. Consequently, cancer cells take more NMe_2_NBS than normal cells.

### 2.5. Intracellular Localization

Cationic phenothiazinium derivatives are reported to be sequestrated in lysosomes due to their high accumulation (as seen in previous studies) [14]. Therefore, we performed lysosomal colocalization of the three PSs with the commercial LysoTracker Green DND 26 in HepG2 cells by confocal laser scanning microscopy. As shown in Figure 2a, the red fluorescence of NH_2_NBS and EtNBS displayed better overlap with the green fluorescence of the commercial LysoTracker than NMe_2_NBS with Pearson’s correlation coefficients (PCC) of 0.76, 0.77, and 0.55, respectively. Owing to the lipophilic cations typically targeting mitochondria, we examined the co-localization of compounds with mitochondria. Surprisingly, Figure 2b shows that two methyl groups that modified NMe_2_NBS (PCC = 0.67) possessed stronger mitochondrial targeting abilities than NH_2_NBS (PCC = 0.14) and EtNBS (PCC = 0.29). Other subcellular distributions of three PSs were measured by commercial organelle trackers (Appendix A). These results confirm that NH_2_NBS and EtNBS are preferentially localized in lysosomes and NMe_2_NBS is accumulated in mitochondria and lysosomes.

### 2.6. Intracellular ROS Generation

To our knowledge, ROS generation plays a key role in PDT-induced cell death [21]. Therefore, we investigated three NIR cationic photosensitizers photoinduced by ROS generation under both normoxic and hypoxic conditions in HepG2 cells by confocal imaging. We used 2,7-dichlorodihydrofluorescein diacetate (DCFH-DA) for detecting intracellular ROS generation. The DCFH-DA with almost no fluorescence can be hydrolyzed by intracellular esterases and then oxidized by ROS to generate DCF with strong green fluorescence, which is used for evaluating oxidative stress in living cells [22]. As shown in Figure 3a, under both normoxic and anoxic conditions, all three compounds can significantly increase the fluorescence of DCFH-DA after 10 min of 635 nm laser irradiation in the order of NH_2_NBS > EtNBS > NMe_2_NBS. The results demonstrated that all PSs were verified as efficient photoinduced-ROS generators in living cancer cells both in normoxia and hypoxia. Singlet oxygen production was confirmed by the singlet oxygen sensor green (SOSG) staining assay (Figure 3b). Three PSs can produce ^1^O_2_ under NIR light irradiation only in a normoxic environment. The production of singlet oxygen by these PSs is dependent on oxygen. Simultaneously, we employed dihydroethidium (DHE) as the indicator to detect the intracellular O_2_^−•^ radical in HepG2 cells [17]. Compared with the control and dark groups, the light groups displayed obvious red fluorescence in normoxia and hypoxia (Figure 3c). This finding indicates that all PSs are excellent O_2_^−•^ radical generators in HepG2 cells, which can overcome the hypoxic environment during PDT treatment.

### 2.7. Intracellular Toxicity Studies

Given the efficient intracellular ROS production, we evaluated the photodynamic anticancer activities of three NIR cationic photosensitizers by the methyl thiazolyl tetrazolium (MTT) assay in normoxia and hypoxia. The photocytotoxicity and dark toxicity of NH_2_NBS, EtNBS, and NMe_2_NBS were determined in human hepatoma cells (HepG2). The complete IC_50_ (half maximal inhibitory concentration) values in different conditions are summarized in Table 1. All PSs exhibited high photocytotoxicity with IC_50_ values of 0.045, 0.072, and 0.287 μM in normoxia and 0.068, 0.104, and 0.447 μM in hypoxia. However, NH_2_NBS (IC_50_ = 2.587 μM) and EtNBS (IC_50_ = 4.874 μM), which were found to be more toxic than the two methyl groups, modified NMe_2_NBS (IC_50_ = 74.557 μM). The phototoxicity and dark toxicity decreased gradually with different alkyl modifications in the photosensitizer structures. Among them, the PDT efficiency of NMe_2_NBS was superior to NH_2_NBS and EtNBS, with high PI values. Additionally, we determined the cytotoxicity of three PSs in the normal human hepatocyte L02 cell line (Appendix A). NH_2_NBS and EtNBS are more toxic-to-normal cells under NIR irradiation. However, the photocytotoxicity of NMe_2_NBS is four times less in normal cells than in cancer cells. This may be because the uptake rate of NMe_2_NBS in normal cells is lower than that in cancer cells (Appendix A). Consequently, NMe_2_NBS has better selectivity in killing cancer cells than NH_2_NBS and EtNBS.

### 2.8. Cytotoxicity Mechanism Studies

As a result of the different cytotoxicity types between NH_2_NBS, EtNBS, and NMe_2_NBS, we have to explore their photo-cytotoxic mechanisms against cancer cells. As there are three PSs localized in lysosomes, we verified PDT-mediated lysosome disruption via acridine orange (AO)-staining in HepG2 cells (Figure 4a and Appendix A). The fluorescence of AO is concentration-dependent, emitting red fluorescence due to aggregation at high concentrations (e.g., in the lysosome) and green fluorescence at low concentrations (e.g., in the cytoplasm) [23]. Thus, AO is typically used as a probe for lysosomal membrane permeabilization (LMP) and a lysosomal integrity indicator. In the dark groups or PS-free groups, red fluorescence from AO was observed, implying the lysosomes were intact. However, in the light group, three PSs caused red fluorescence to disappear under normoxia, indicating that the induced LMP was enhanced and the lysosomes were severely damaged. In hypoxic conditions, red fluorescence of AO under irradiated NMe_2_NBS is still displayed. Therefore, the degree of photoinduced LMP and lysosomal destruction is NH_2_NBS > EtNBS > NMe_2_NBS. The dark toxicity might be derived from lysosome damage. We co-incubated cells with three PSs (1 μM) for 24 h in the dark and then with AO for 30 min before CLSM. Figure 4a shows that the red fluorescence of AO decreased in the NH_2_NBS and EtNBS groups. The results revealed that under dark conditions, both NH_2_NBS and EtNBS could induce lysosomal damage, but NMe_2_NBS could not. This trend is consistent with dark toxicity. Since NMe_2_NBS had a higher photocytotoxicity index (PI) value and better biocompatibility than NH_2_NBS and EtNBS, this compound was selected for subsequent in vivo photodynamic therapy. 

We speculated that the damaged lysosome might release the sequestered PSs into other organelles. As shown in Figure 4b, the intracellular positions of NH_2_NBS and EtNBS were significantly changed under light irradiation. We assumed that the released cationic PSs would enter into the mitochondria due to their lipophilic cationic characterization. Subsequently, the mitochondrial localizations of three PSs were conducted after NIR light irradiation. As expected, the PCC of NH_2_NBS (0.86) and EtNBS (0.79) significantly improved compared to before (Figure 4c). Generally, photoinduced ROS in mitochondria will disturb the mitochondrial membrane potential (MMP), which is the key indicator of the mitochondrial state. After 635 nm of light irradiation, using JC-1 as a fluorescent probe [24,25], MMP was determined by confocal imaging in HepG2 cells (Figure 4d). In the PS-free groups, we can observe weak green fluorescence and intensive red fluorescence, elucidating a high MMP value. In the NH_2_NBS, EtNBS, and NMe_2_NBS groups, there is intensive green fluorescence and weak red emission, indicating an obvious MMP decrease. Therefore, these photosensitizers were released from lysosomes and entered into mitochondria, which damaged the lysosome and mitochondria after exposure to light.

Following lysosomal or mitochondrial damage, photosensitizers may be present in the cytoplasm. In the cytoplasm, the nicotinamide adenine dinucleotide 2′-phosphate (NADPH) is a very important coenzyme involved in many redox reactions in biosynthesis and metabolism [26,27]. Moreover, it acts as a reducing agent to maintain intracellular redox balance by cycling between reducing (NADPH) and oxidizing (NADP) forms. Thus, we investigated the photooxidation of NH_2_NBS, EtNBS, and NMe_2_NBS with NADPH both in the solution and in cancer cells (Figure 5). In the PBS solution, the absorption band at 330 nm belongs to the characteristic absorption band of NADPH. In the PS-free group, there are no changes in the absorption spectrum of NADPH after ten minutes of 635 nm of light irradiation. On the contrary, NH_2_NBS, EtNBS, and NMe_2_NBS gradually reduced the absorption of NADPH under irradiation for ten minutes Figure 5a and Appendix A). This finding suggests that these PSs can photo-oxide NADPH, and the capability of photo-oxidation is NH_2_NBS > EtNBS > NMe_2_NBS. At the same time, we determined the level of NADPH in HepG2 cells with or without three PS incubations in dark or light conditions (Figure 5b). The NADPH level was about 30 nmol mg^−1^ protein in HepG2 cells with PS-free or dark PS incubation. After ten min of irradiation, the intracellular NADPH with NH_2_NBS, EtNBS, and NMe_2_NBS incubations decreased distinctively. Therefore, these PSs oxidized cellular NADPH under irradiation with the trend of NH_2_NBS > EtNBS > NMe_2_NBS.

There are various pathways to PDT-mediated cell death, and apoptosis is the most common one. The photoinduced cell apoptosis by NH_2_NBS, EtNBS, and NMe_2_NBS was determined with Annexin V-FITC as an indicator. From Appendix A, green fluorescence was observed in the light group, suggesting that light-induced cancer cell death is through the apoptosis pathway. The results of flow cytometry also confirmed photoinduced apoptosis and the percentage of apoptosis increased with the increasing PS concentration (Appendix A). The above results demonstrate that the photoinduced ROS of three PSs synergistically disrupted lysosomes and mitochondria, reduced NADPH, and induced cell apoptosis.

### 2.9. In Vivo Photodynamic Antitumor Studies

Benefiting from the good performance of NMe_2_NBS (i.e., in negligible dark toxicity, moderating phototoxicity, and being selective against cancer cells), we explored its photodynamic antitumor efficacy in HepG2 tumor-bearing female nude mice models. In vivo, NIR fluorescence imaging was performed after intratumorally injected NMe_2_NBS (50 μM, 100 uL) via a small animal in vivo imaging system. As shown in Figure 6, the tumor tissue was visualized and distinguished from surrounding healthy tissue. The fluorescence intensity reached its maximum at 4 h and persisted for longer periods of time. After 24 h of NMe_2_NBS injection, the mice were sacrificed and their major organs (hearts, livers, spleens, lungs, and kidneys) and tumor tissues were removed for imaging. The tumor tissue displayed higher fluorescence than other organs, implying that NMe_2_NBS persisted in tumors and did not spread to other organs.

Intratumoral injection is a practical and clinical cancer treatment strategy that enables greater drug diffusion within tumors, minimizes unwanted biodistribution, and reduces systemic toxicity and side effects [28]. Herein, we used intratumoral administration to investigate the tumor-suppressive effect of the NMe_2_NBS. After four weeks of treatment, the tumor volumes of the control group and the dark group increased dramatically. In comparison, the tumors of the PDT group were significantly suppressed or even ablated (Figure 7). Finally, the mice were sacrificed and tumors were removed and weighed. The average tumor weights of the PDT group were lower than those in the control group and dark group, exhibiting 89% tumor growth inhibition. It is important to mention that the body weights of mice did not change throughout the entire therapeutic period. Finally, the major organs (hearts, livers, spleens, lungs, and kidneys) and tumor tissues were obtained and then analyzed histologically by hematoxylin and eosin (H&E) staining (Appendix A). The H&E staining of tumors in the control and dark groups displayed negligible necrosis, indicating that NMe_2_NBS has negligible dark toxicity to tumor cells. In contrast, the slide of the PDT group showed serious cell damage, implying that NMe_2_NBS has outstanding antitumor efficacy under NIR light irradiation.

Fortunately, in the stained organ slide images, no obvious pathological abnormalities were observed in any of the groups. These results demonstrate that NMe_2_NBS is an efficient PDT agent with excellent biocompatibility and safety in vivo.

## 3. Materials and Methods

All reagents were purchased from the Aladdin company, Macklin company, and General Regent company. Moreover, all reagents were used directly without any treatment. The characterizations of ^1^HNMR and ^13^CNMR spectra were recorded from the Bruker Avence III 400 MHz spectrometer (Bruker, Ettlingen, Germany) with TMS as the internal standard. We used an Agilent 6540 Q-TOF HPLC-MS spectrometer (Agilent, CA, USA) to characterize the high-resolution mass spectra (HRMS). For fluorescent spectroscopic studies, the HORIBA FluoroMax-4 spectrofluorometer (Horiba, CA, USA) was utilized. Similarly, a Perkine-Elmer Lambda 1050+ spectrophotometer (Perkine Elmer, MA, USA) was utilized for recording the absorption spectra. A Sartorius PB-10 m with a combined glass calomel electrode was used for recording the pH values. All of these spectroscopic measurements were conducted at room temperature. A Zeiss confocal laser scanning microscope LSM 710 (Carl Zeiss, Jena, Germany) was used for cell imaging

### 3.1. Synthesis and Characterization

Synthesis of 2-amino-5-diethyl-aminophenylthiosulfuric acid (Bunte salt) [16]. *N*,*N*,diethyl-p-phenylenediamine (1 g, 6.09 mmol) that was dissolved in 5 mL of methanol was added to a stirred solution of aluminum sulfate (3.125 g, 9.12 mmol) in water (20 mL). Sodium thiosulfate (2.21 g, 14 mmol) followed by zinc chloride (0.872 g, 6.39 mmol) were added to the reaction mixture. The reaction mixture was cooled in an ice bath and an aqueous solution of potassium dichromate (0.49 g, 1.68 mmol) (5 mL) was added slowly over 20 min. The solution was stirred in an ice bath for 2 h, filtered, and the observed thick precipitate was rinsed with water and ether: ethanol (1:1). The crude product was refluxed in methanol (20 mL) and filtered to give a dark gray solid. The product‘s mass peaks were 275 and 298. This product was used without further purification or characterization.

Bunte salt (276 mg, 1.0 mmol), naphthyl derivative (286 mg, 2.0 mmol), and 25 mL of methanol were placed in a round bottom flask equipped with a magnetic stirrer and reflux condenser. This reaction mixture was heated to reflux temperature. Silver carbonate (606 mg, 2.20 mmol) was slowly added to the refluxing reaction mixture and an intense color change was observed within 5 min of the complete addition. After heating for 30 min, the reaction flask was cooled to room temperature. The reaction mixture was filtered through a pad of Celite. Then, KPF_6_ aqueous was added to the mixture. After stirring for 2 h, the reaction mixture was evaporated to obtain a dark blue crude product. The crude product was redissolved in 25 mL of dichloromethane, washed with saturated sodium carbonate solution, and dried over sodium sulfate. The NH_2_NBS product was obtained as a dark blue solid and purified by column chromatography using 1–10% methanol in dichloromethane as an eluent.

^1^H NMR (400 MHz, DMSO-*d*_6_) δ 9.52 (s, 2H), 8.91 (d, *J* = 8.0 Hz, 1H), 8.44–8.34 (m, 1H), 7.99–7.78 (m, 3H), 7.32 (s, 2H), 7.16 (s, 1H), 3.63 (d, *J* = 7.1 Hz, 4H), 1.24 (t, *J* = 7.0 Hz, 6H).

^13^C NMR (101 MHz, DMSO-*d*_6_) δ 156.13, 150.77, 150.74, 138.72, 136.79, 133.76, 133.73, 133.27, 131.79, 131.77, 131.69, 130.73, 129.40, 124.62, 123.78, 123.35, 117.11, 106.11, 105.26, 45.07, 12.54.

HRMS: (positive mode, *m*/*z*) Calcd. 334.1378, found 334.1372 for [M]^+^.

EtNBS was obtained by the same procedure as NH_2_NBS.

^1^H NMR (400 MHz, DMSO-*d*_6_) δ 9.71 (s, 1H), 8.95–8.86 (m, 1H), 8.43 (t, *J* = 8.6 Hz, 1H), 7.96–7.87 (m, 2H), 7.85 (t, *J* = 6.9 Hz, 1H), 7.46 (d, *J* = 15.8 Hz, 1H), 7.39–7.31 (m, 2H), 3.73 (q, *J* = 5.8, 4.8 Hz, 2H), 3.66 (q, *J* = 7.2 Hz, 4H), 1.37 (t, *J* = 7.2 Hz, 3H), 1.24 (t, *J* = 7.0 Hz, 6H).

^13^C NMR (101 MHz, DMSO-*d*_6_) δ 152.67, 150.73, 139.65, 136.73, 133.40, 133.05, 133.00, 131.99, 131.95, 131.22, 131.09, 129.40, 124.61, 123.96, 123.92, 122.99, 117.27, 105.25, 102.76, 45.03, 38.93, 13.87, 12.57.

HRMS: (Positive mode, *m*/*z*) Calcd. 362.1685, found 362.1688 for [M]^+^.

NMe_2_NBS was synthesized by the same procedure as NH_2_NBS.

^1^H NMR (400 MHz, DMSO-*d*_6_) δ 8.89 (dd, *J* = 38.9, 8.1 Hz, 1H), 8.24 (dd, *J* = 20.9, 8.3 Hz, 1H), 7.99–7.82 (m, 2H), 7.78 (dd, *J* = 13.2, 6.2 Hz, 1H), 7.54 (d, *J* = 41.9 Hz, 1H), 7.39 (d, *J* = 29.0 Hz, 2H), 3.67 (t, *J* = 7.0 Hz, 4H), 3.55 (d, *J* = 9.4 Hz, 6H), 1.25 (t, *J* = 7.0 Hz, 6H).

^13^C NMR (101 MHz, DMSO-*d*_6_) δ 157.51, 150.92, 150.88, 137.07, 136.86, 134.49, 132.39, 131.58, 130.68, 128.30, 127.76, 125.03, 124.73, 117.66, 108.22, 105.50, 45.22, 45.20, 45.13, 12.60.

HRMS: (Positive mode, *m*/*z*) Calcd. 362.1685, found 362.1688 for [M]^+^.

### 3.2. Photophysical Property Studies

The stock solutions (10 mM) of NH_2_NBS, EtNBS, and NMe_2_NBS were prepared by dissolving in DMSO. All compounds were diluted with different solvents to a concentration of 10 μM for absorption emission spectra tests. The fluorescence quantum yields of three PSs were determined by the following equation:Φ_F(sam)_ = Φ_F(ref)_ (F_sam_/F_ref_) (A_ref_/A_sam_) (η^2^_sample_/η^2^_ref_),(1)
where F is the integrated fluorescence area, A is the absorption value at the excitation wavelength, and η is the refractive index. Nile Blue in EtOH was selected as the reference (Φ_F(ref)_ = 0.27).

### 3.3. ROS Detection

#### 3.3.1. Singlet Oxygen Generation

Using 1,3-diphenylisobenzofuran (DPBF) as the singlet oxygen scavenger and monitoring the changes in absorbance at 410 nm, the singlet oxygen quantum yields of the three PSs were determined by the relative measurement method in MeOH. The calculation formula is as follows:Φ_△sample_ = Φ_△ref_ (K_sam_/K_ref_) (F_ref_/F_sam_),(2)

Φ_△ref_ is the singlet oxygen quantum yield of reference (MB, Φ_△_ = 0.50 in methanol) and K_sam_ and K_ref_ are the slopes of change in the maximum absorption of DPBF versus the irradiation time. F is the absorption correction factor, which is expressed by F = 1–10^−OD^, where OD is the absorption value at 635 nm.

#### 3.3.2. Superoxide Radical Generation

The commercial O_2_^−•^ fluorescent probe dihydrorhodamine 123 (DHR123) determined the superoxide radical production of three photosensitizers in the PBS solution. The concentration of DHR 123 and three PSs was 10 μM for the emission spectra test after 635 nm of light irradiation at different times.

### 3.4. Cell Culture

HepG2 cells were cultured in DMEM medium with 10% fetal bovine serum, 100 units/mL of penicillin, and 50 units/mL of streptomycin. The condition of the incubator was the atmosphere containing 5% CO_2_ and 95% air at 37 °C. In addition, in order to mimic the hypoxic condition, cells were sealed in an anaerobic airbag and other options were the same as normoxia.

### 3.5. Cellular Uptake

The HepG2 cells and L02 cells were seeded on a sterile six-well plate and cultured in a DMEM medium with 10% fetal bovine serum in the atmosphere containing 5% CO_2_ and 95% air at 37 °C. After 24 h of incubation in the dark, the medium was replaced with 2 mL of phenol red-free HBSS containing 10 μM of PS, and three wells for each PS. After incubating HepG2 cells with 10 μM of PS at 37 °C in a phenol red-free HBSS solution for 20, 40, and 60 min, the cells grown on culture plates were washed three times with cold HBSS, harvested, placed in tubes, and centrifugated. The three PSs could be extracted from cells with 1.005 mL of solvent mixture (methanol:chloroform:acetic acid, 100:100:1). The cellular content of PSs could be calculated by absorbance values

### 3.6. Intracellular Localization

The HepG2 cells were seeded in the glass bottom cell culture dishes and grew overnight. After removing the medium, HepG2 cells were incubated with NH_2_NBS, EtNBS, and NMe_2_NBS (100 nM) at 37 °C for 1 h, and then replaced with DMEM medium containing LysoTracker Green DND 26 (75 nM) and Mito-Tracker Green (200 nM), and incubated for another 30 min. After three washes with PBS, the cells were immediately imaged with Zeiss LSM 710. The excitation wavelength for LysoTracker Green DND 26 and Mito-Tracker Green was 488 nm and the imaging band path was 500–550 nm. The excitation wavelength for NH_2_NBS, EtNBS, and NMe_2_NBS was 633 nm and the imaging band path was 650–750 nm

### 3.7. Intracellular ROS Detection

Intracellular ROS generation induced by NH_2_NBS, EtNBS, and NMe_2_NBS was detected by 2,7-dichlorodihydrofluorescein diacetate (DCFH-DA) as an indicator. Singlet oxygen production was confirmed by the singlet oxygen sensor green (SOSG) staining assay. Dihydroethidium (DHE) was used as the indicator to detect the intracellular O_2_^−•^ radical in HepG2 cells. HepG2 cells were incubated with three PSs (1 μM) for 1 h, then the medium was replaced with DCFH-DA (10 μM), SOSG (5 μM), and DHE (5 μM), and incubated for 30 min, respectively. After that, the cells were irradiated with 635 nm laser irradiation for 10 min at a power density of 20 mW/cm^2^. Then the cells were immediately imaged with Zeiss LSM 710. The excitation wavelength for DCFH-DA, SOSG, and DHE was 488 nm, and the emission wavelengths were collected from 500 to 550 nm for DCFH-DA and SOSG, and 570 to 630 nm for DHE.

### 3.8. Cell Viability Assay

The cytotoxic studies of NH_2_NBS, EtNBS, and NMe_2_NBS against HepG2 and L02 cells were detected by the methyl thiazolyl tetrazolium (MTT) assay. HepG2 and L02 cells were seeded in 96-well plates at 4500 cells per well, grew overnight in the incubator, and spent another 2 h under normoxia (21% O_2_) and hypoxia (below 1% O_2_) atmospheres. Then the medium was replaced by 200 μL of PS with varying concentrations in each well and incubated for 1 h in the dark under normoxia and hypoxia. For the light groups, the cells were irradiated with 635 nm laser irradiation for 10 min at a power density of 20 mW/cm^2^. For the hypoxia group, all of the steps were the same and the anaerobic airbag was removed after irradiation. For the dark group, all of the steps were the same except for irradiation. Cells were further cultured for 24 h. Then 40 μL of MTT solution (2.5 mg/mL) was added to each well. After 4 h, the medium was carefully removed from the wells, 150 μL of DMSO was added to each well, and the purple formazan was fully dissolved by shaking the plate for 15 min. The absorbance of purple formazan in each well was measured at 490 nm by a microplate reader. These independent experiments were repeated at least three times. The cell viability was calculated by the following formula:Cell viability (%) = (ODps−ODblank control/ODcontrol−ODblank control) × 100%,(3)

### 3.9. Lysosome Disruption Analysis

The lysosomal damage was assessed by acridine orange (AO) staining. HepG2 cells were seeded in the glass bottom confocal dishes and grew overnight. After removing the medium, HepG2 cells were incubated with NH_2_NBS, EtNBS, and NMe_2_NBS (100 nM) at 37 °C for 1 h, and then replaced with DMEM medium containing AO (5 μM), incubated for another 30 min, respectively. After that, the cells were irradiated with 635 nm laser irradiation for 10 min at a power density of 20 mW/cm^2^. For the hypoxia group, all of the steps were the same, the anaerobic airbag was removed after irradiation. Then the cells were immediately imaged with Zeiss LSM 710. The excitation wavelength for AO was 488 nm, and the emission wavelengths were collected from 500 to 550 nm for the green channel and 610–640 nm for the red channel.

### 3.10. Mitochondrial Membrane Potential Assay

Mitochondrial membrane potential (MMP) was determined by using JC-1 as a fluorescent probe. HepG2 cells were seeded in the glass bottom confocal dishes and grew overnight. After removing the medium, HepG2 cells were incubated with NH_2_NBS, EtNBS, and NMe_2_NBS (100 nM) at 37 °C for 1 h, and then replaced with JC-1 (Beyotime Biotechnology), respectively. After that, the cells were irradiated with 635 nm laser irradiation for 10 min at a power density of 20 mW/cm^2^. For the hypoxia group, all of the steps were the same; the anaerobic airbag was removed after irradiation. Then the cells were immediately imaged with Zeiss LSM 710. The excitation wavelength for JC-1 was 488 nm and the emission wavelengths were collected from 530 to 560 nm for the green channel; the excitation wavelength for the red channel was 543 nm, and the emission wavelengths were collected from 580 to 640 nm.

### 3.11. Intracellular NADPH Detection

The HepG2 cells were seeded in a sterile six-well plate and cultured overnight. After 24 h, the level of NADPH was determined by the CheKine^TM^ Coenzyme II NADPH Assay Kit (Abbkin, Wuhan, China) and the procedure followed the manufacturer’s protocol.

### 3.12. Flow Cytometry

The Annexin V-FITC Apoptosis Detection Kit (Beyotime Biotect Inc, Shanghai, China) was used to detect that the three PSs mediated photoinduced cell death. The HepG2 cells were cultured in a sterile six-well plate and incubated with NH_2_NBS (0.04, 0.08, and 0.16 μM), EtNBSN (0.09, 0.18, and 0.36 μM), and NMe_2_NBS (0, 0.5 μM) at 37 °C for 1 h in the dark. Then, the cells were irradiated with 635 nm laser irradiation for 10 min at a power density of 20 mW/cm^2^. After 12 h, cells were stained with Annexin V-FITC (following the manufacturer’s protocol) and were then subjected to the flow cytometric analysis.

### 3.13. In Vivo Fluorescence Imaging and PDT Evaluation

The female BALB/c nude mice (4–5 weeks) were obtained from Qing Long Shan Animal Breeding Farm, Jiangning District, Nanjing, Jiangsu province, China. Then, 5 × 10^6^ HepG2 cells were injected subcutaneously at selected positions to establish the tumor model. When the tumor volume reached 200 mm^3^, mice were used for imaging and the PDT evaluation. 

In vivo, NIR fluorescence imaging was performed after intratumorally injecting NMe_2_NBS (50 μM, 100 uL) via a small animal in vivo imaging system. After 24 h of injecting NMe_2_NBS, the mice were sacrificed and their major organs (hearts, livers, spleens, lungs, and kidneys) and tumor tissues were removed for imaging. The excitation wavelength was 660 nm and the emission wavelength was collected at 710 nm. 

The mice were randomly divided into three groups, i.e., the control group (PBS + light), the dark group (only NMe_2_NBS), and the PDT group (NMe_2_NBS + light). PBS (100 uL) and NMe_2_NBS (50 μM, 100 uL) were injected intratumorally into the tumor-bearing mice of different groups, respectively. The subsequent treatments were initiated by 635 nm laser irradiation (100 mW, 10 min) at 4 h post-injection, and treatments were performed every two days. The body weights and tumor volumes were recorded every two days to observe the therapeutic effects.

### 3.14. In Vivo Histological Assay

The in vivo histological analysis was achieved by the H&E section’s histological analysis. After four weeks of treatment, the mice were sacrificed and their major organs (hearts, livers, spleens, lungs, and kidneys) and tumor tissue were obtained, and then analyzed histologically by hematoxylin and eosin staining

## 4. Conclusions

In summary, we synthesized and characterized cationic NIR phenothiazinium NH_2_NBS, EtNBS, and NMe_2_NBS as both type I and type II photosensitizers for PDT. All PSs showed distinct capabilities of generating O_2_^−•^ in vitro and exhibited high photocytotoxicity under both normoxic and hypoxic conditions. Comprehensive investigations were performed to understand the cytotoxic mechanism. Our findings revealed that the photocytotoxicity differences were related to their subcellular locations caused by different alkylation modifications. NH_2_NBS and ethyl-modified EtNBS were mainly targeted in the lysosome, while two methyl-modified NMe_2_NBS showed higher accumulation in mitochondria than in lysosomes. After irradiation, NH_2_NBS and EtNBS were released from the lysosomes and relocated into the mitochondria. Hence, photoinduced ROS of three PSs synergistically disrupted the lysosomes and mitochondria (stepwise or by separately enhancing the lysosomal membrane permeability and reducing the mitochondrial membrane potential). Additionally, these PSs can break intracellular redox balance by oxidizing cellular NADPH under irradiation. Furthermore, two methyl-modified NMe_2_NBS showed excellent photodynamic antitumor efficacy without apparent side effects in vivo. We believe that this work provides a powerful methodology to further explore the relationship between the structural modification of benzophenothiazine and the mechanism of cytotoxicity, which would be of great importance for their clinical applications in future cancer therapy.

## Data Availability

The data supporting the reported results can be found within the article and its Appendix A.

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
