# Peer review of "Manipulating the Subcellular Localization and Anticancer Effects of Benzophenothiaziniums by Minor Alterations of N-Alkylation"

_molecules, 2023, doi:10.3390/molecules28041714_

Round 1
Reviewer 1 Report
This paper describes Manipulating the Subcellular Localization and Anti-cancer Effect of Benzophenothiaziniums by Minor Alteration of N-Alkylation. The differences in behavior for the three derivatives according to their subcellular location, mechanism of action and structure were interesting. However, some fixes have been identified and should be fixed.
1. This study proposes the synthesis of three types of benzophenothiazine derivatives through minor N-alkyl alteration. Then, if a reasonable basis for using minor N-alkyl alteration in the structure underlying these derivatives is presented, the understanding of why the three types of derivatives are used will increase.
2. Is there a reason why only three types of benzophenothiazine derivatives were tested without using unmodified benzophenothiazine (i.e., benzophenothiazine without minor N-alkyl alteration in this study) as a control group?
3. Excellent explanation of data composition and results. However, no supporting discussion of the results has been conducted. (ex) Why do cancer cells have high P/S uptake efficiency?) => We suggest adding a discussion of the rationale for deriving the outcome data.
4. Figure S3 legend needs to be modified (a,b notation is wrong)
5. After supplementary data S10, only figure legends are specified and explanations are not provided. We propose adding interpretations to supplementary data after S10
6. In the explanation of result 2.6, the concept of normoxic and hypoxic is not explained before, but immediately appears in the explanation of the result, so it is difficult to naturally interpret the result. If the reason for why the comparison between normoxia and hypoxia should be explained, it would help interpret the results.
+ It would be nice if singlet oxygen was also explained earlier as above why it is an important concept in this paper.
7. Information on the analysis instrument must be entered. e.g. FACS (country, region, etc.)
Author Response
Response to Reviewer 1’s Comments
Comments: This paper describes Manipulating the Subcellular Localization and Anti-cancer Effect of Benzophenothiaziniums by Minor Alteration of N-Alkylation. The differences in behavior for the three derivatives according to their subcellular location, mechanism of action and structure were interesting. However, some fixes have been identified and should be fixed.
Response: Thanks a lot for your patience and very positive comments and we have revised the manuscript according to your advice.
Question 1: This study proposes the synthesis of three types of benzophenothiazine derivatives through minor N-alkyl alteration. Then, if a reasonable basis for using minor N-alkyl alteration in the structure underlying these derivatives is presented, the understanding of why the three types of derivatives are used will increase.
Answer 1: Thank you for your suggestion. We use N-alkyl alteration in the structure to maintain the basic photophysical properties to be similar to its derivatives. Because benzophenothiazine based photosensitizers has a twisted intramolecular charge transfer between the cationic chromophore and amino auxochrome. The twisted intramolecular charge transfer has been proposed as a possible relaxation route from the excited state to the ground state. We speculated that charge separation and non-radiative relaxation maybe occurs most strongly with N-alkyl alteration. To better understand, we are adding more descriptions in revised manuscript (Line 67).
Question 2: Is there a reason why only three types of benzophenothiazine derivatives were tested without using unmodified benzophenothiazine (i.e., benzophenothiazine without minor N-alkyl alteration in this study) as a control group?
Answer 2: Thank you for your suggestion. The compound NH2NBS has been extensively studied for many years, so it is the most suitable as a control group. The N-alkyl is crucial for benzophenothiazine based photosensitizers. However, we only made some slight modification on the N-alkyl moiety to see if minor alteration could cause significant change on photophysical properties and PDT effect. Significant changes on the N-alkyl moiety might also greatly affect the lipophilic properties and could make things more complicated. The related investigation could be interesting and is currently undergoing in our lab, which will be reported in the future.
Question 3: Excellent explanation of data composition and results. However, no supporting discussion of the results has been conducted. (ex) Why do cancer cells have high P/S uptake efficiency?) We suggest adding a discussion of the rationale for deriving the outcome data.
Answer 3: Thank you for your suggestion. We have added some discussion of the rationale for deriving the outcome data in part 2.4 (Line 134 and line 136).
Question 4: Figure S3 legend needs to be modified (a,b notation is wrong)
Answer 4: We are really sorry for this mistake and thanks a lot for your kind reminding. We modified the Figure S3 legend.
Question 5: After supplementary data S10, only figure legends are specified and explanations are not provided. We propose adding interpretations to supplementary data after S10.
Answer 5: Thanks to reviewer for reminder. We have added interpretations to supplementary data after S10 in revised manuscript (Part 2.1, line 91).
Question 6: In the explanation of result 2.6, the concept of normoxic and hypoxic is not explained before, but immediately appears in the explanation of the result, so it is difficult to naturally interpret the result. If the reason for why the comparison between normoxia and hypoxia should be explained, it would help interpret the results. It would be nice if singlet oxygen was also explained earlier as above why it is an important concept in this paper.
Answer 6: Thanks to reviewer for reminder. We have added description in the introduction section (Line 40).
Question 7: Information on the analysis instrument must be entered. e.g. FACS (country, region, etc.)
Answer 7: Thanks to reviewer for reminder. We have entered information on the analysis instrument (Part 3, line 328).
Reviewer 2 Report
Review Report
The work "Manipulating the Subcellular Localization and Anti-cancer Effect of Benzophenothiaziniums by Minor Alteration of N-Alkylation" by Yanping Wu et al. study the effects on structure-property relationships in a series of synthesized benzophenothiazine derivatives with minor N-alkyl alteration.
The paper adds valuable data and methodology to investigate the structure-property relationship of benzophenothiazine dyes, which is of important for practical application of photodynamic therapy against hypoxia tumor cells.
I recommend to publish it in a present form.
Author Response
Response to Reviewer 2’s Comments
Comments: The work "Manipulating the Subcellular Localization and Anti-cancer Effect of Benzophenothiaziniums by Minor Alteration of N-Alkylation" by Yanping Wu et al. study the effects on structure-property relationships in a series of synthesized benzophenothiazine derivatives with minor N-alkyl alteration. The paper adds valuable data and methodology to investigate the structure-property relationship of benzophenothiazine dyes, which is of important for practical application of photodynamic therapy against hypoxia tumor cells. I recommend to publish it in a present form.
Response: Thanks a lot for your patience and very positive comments.